# Kinins and Their Receptors in Infectious Diseases

**DOI:** 10.3390/ph13090215

**Published:** 2020-08-27

**Authors:** Ana Paula A. Dagnino, Maria M. Campos, Rodrigo B. M. Silva

**Affiliations:** Centro de Pesquisa em Toxicologia e Farmacologia, Escola de Ciências da Saúde e da Vida, Pontifícia Universidade Católica do Rio Grande do Sul, Porto Alegre, RS 90619-900, Brazil; anapaula.dagnino@gmail.com (A.P.A.D.); rodrigo.braccinims@gmail.com (R.B.M.S.)

**Keywords:** bradykinin, B_1_ receptors, B_2_ receptors, infection, therapeutics

## Abstract

Kinins and their receptors have been implicated in a series of pathological alterations, representing attractive pharmacological targets for several diseases. The present review article aims to discuss the role of the kinin system in infectious diseases. Literature data provides compelling evidence about the participation of kinins in infections caused by diverse agents, including viral, bacterial, fungal, protozoan, and helminth-related ills. It is tempting to propose that modulation of kinin actions and production might be an adjuvant strategy for management of infection-related complications.

## 1. Introduction

Bradykinin (BK) and related kinins are a family of small active peptides implicated in a series of biological effects. The nonapeptide BK is cleaved from the high molecular weight kininogen (HMWK), via activation of plasma kallikrein. BK generation in plasma takes part in the intrinsic coagulation pathway activation, involving the interaction of Factor XII (FXII), prekallikrein (PPK) and Factor XI (FXI) with HMWK, leading to prothrombotic and inflammatory effects [1,2]. Alternatively, kallidin (Lys-BK) is a decapeptide formed by the action of tissue kallikrein on low molecular weight kininogen (LMWK) precursor. In some circumstances, Lys-BK can be converted into BK following the cleavage of the amino-terminal lysine, by the action of plasmatic aminopeptidases. Both Lys-BK and BK are short-acting mediators that are rapidly degraded into inactive fragments by kininase 2, also called angiotensin-converting enzyme (ACE). Under certain circumstances, such as in an inflammatory milieu, there is an increased affinity of kininase 1 for kinins. In this context, this enzyme converts Lys-BK and BK into the active kinins, namely Lys-des-Arg^9^-BK and des-Arg^9^-BK, respectively [3,4,5,6]. Interestingly, ACE and its isoform ACE2 are zinc metallopeptidases involved in the metabolism of vasoactive peptides, having angiotensin- and kinin-family members as their main substrates, displaying a relevant role in a series of pathophysiological processes [7].

Predominantly, the effects of kinins are mediated by the activation of two G protein-coupled receptors, denoted as B_1_ and B_2_. These receptors have been extensively characterized in several animal species and humans, showing an identity of about 35%, with small variations across species. Both receptors share common signaling pathways, but their pattern of expression is distinctive. The B_2_ receptor is defined as a housekeeping molecule, being constitutively expressed by most cells at peripheral and central levels, such as vascular and non-vascular smooth muscle cells, epithelial and immune cells, besides neurons and glial cells. Conversely, despite some exceptions, the B_1_ receptor displays a low physiological expression, being upregulated under a variety of stressful stimuli, such as the exposure to infectious agents. Another difference refers to the affinity for the natural agonists: the B_2_ receptor has a high affinity for Lys-BK and BK, while the B_1_ receptor preferentially binds to Lys-des-Arg^9^-BK and des-Arg^9^-BK. Irrespective of these differences, both B_1_ and B_2_ receptors have been implicated in a series of responses, including the regulation of blood vessel tonus, inflammatory changes, and pain processing mechanisms [3,6,8,9,10].

The advances in the comprehension of the pathophysiological roles of kinins and their receptors have been greatly favored by development of peptide and non-peptide selective receptor ligands by chemical synthesis [4,8,9,11,12,13]. Moreover, the modulation of kallikreins and kininases has also been explored for therapeutic purposes [14]. Interestingly, the selective pseudopeptide B_2_ receptor antagonist HOE-140 (Icatibant) has been approved in the Americas and Europe, for managing acute episodes of hereditary angioedema, under the commercial name Firazyr [15]. Additionally, the therapeutic benefits of ACE inhibitors, such as captopril, also rely on the modulation of kinins and their biological effects [16]. Over the last four decades, the role of either kinin receptor and other components of the kinin system has been described in a series of infectious conditions, such as sepsis, schistosomiasis, leishmaniosis, Chagas Disease, candidiasis, tuberculosis, malaria, and it has been recently correlated with Covid-19 infection [17,18,19,20,21,22]. Considering the abovementioned evidence, the present review article aims to discuss the relevance of the kinin system, mainly of kinin receptors, as pharmacological targets for management of infectious diseases, attempting to cover the implication of kinins in the progression of infection-related illnesses in humans.

## 2. Bacterial Infections and Kinins

According to the World Health Organization [23] communicable diseases caused by bacterial infections stand out among the top ten causes of deaths worldwide. In 2016, lower tract respiratory infections, diarrhea and tuberculosis accounted for 6 million deaths globally. The burden of bacterial infectious diseases is especially relevant in low-income countries, but it has gained more attention due to the rapid advent of antibiotic resistance in the last decades [24]. In this regard, there is a current interest in relation to the pathways implicated in bacterial infectious diseases, mainly focused on the bacteria-host interaction mechanisms. This section will discuss the most relevant data regarding the participation of the kinin family of molecules in bacterial infections. A summary of main findings discussed in this section is provided in Figure 1.

### 2.1. Bacterial Endotoxins and Kinin Responses

Lipopolysaccharide (LPS) is a component of the cell wall of gram-negative bacteria that elicits marked inflammatory responses. Its effects are primarily mediated by the activation of Toll-like receptors (TLR), mainly via the TLR-4-MD-2 complex, in a process dependent on LPS-binding protein (LPB) and CD14 [25]. This set of responses takes part of the host innate immune system activation triggered by pathogen-associated molecular patterns, such as LPS, aimed to suppress the infectious damage [26].

The relationship between LPS responses and the activation of the kinin system has been widely investigated in several experimental paradigms. A classical study conducted by Regoli et al. (1981) demonstrated that strips of large arteries and veins, obtained from rabbits that had been treated with LPS before euthanasia, displayed marked contractile responses to the selective B_1_ receptor agonists, des-Arg^9^-BK and Lys-des-Arg^9^-BK. This evidence provided the basis indicating that infectious stimuli, in addition to trauma responses, were able to promote an upregulation of B_1_ receptors, with variable responses regarding the B_2_ receptors [27]. This notion was extended by studies showing that both B_1_ receptor agonists, des-Arg^9^-BK and Lys-des-Arg^9^-BK, led to hypotensive responses in rabbits pre-treated with sub-lethal doses of LPS, according to assessment of mean blood pressure in comparison with untreated animals [28]. Subsequently, the induction of B_1_ receptors in rabbit isolated aorta preparations was also demonstrated by the in vitro incubation with bacterial LPS, which led to a time- and concentration-dependent increase of contractile responses to des-Arg^9^-BK [29]. The upregulation of B_1_ receptor-mediated contractile responses was also showed in non-vascular preparations, such as the isolated urinary bladder, after the in vitro or in vivo exposure to LPS [30].

As for inflammation models, the intravenous treatment with a low dose of LPS caused an increase of rat paw edema induced by the selective B_1_ receptor agonist des-Arg^9^-BK, whereas the edema formation produced by the selective B_2_ receptor agonist tyrosine^8^-BK was markedly reduced. The changes of kinin receptor-mediated inflammatory responses in LPS-treated animals were sensitive to glucocorticoids and cycloheximide, which suggests an implication of protein synthesis in the opposite balance of B_1_ and B_2_ receptors under inflammatory conditions [31]. Notably, the intravenous administration of LPS also produced an inversion of B_2_/B_1_ receptor-mediated febrile responses: the hyperthermia induced by the central administration of BK and tyrosine^8^-BK was reduced in LPS-treated rats, whereas the fever elicited by des-Arg^9^-BK was noticeably increased [32]. Remarkably, the effects of LPS on B_1_ and B_2_ receptors might be variable depending on the experimental approach. For instance, Passos et al. (2004) showed that local application of LPS into the rat paw resulted in a time-related increase of des-Arg^9^-BK-induced edema formation, preceded by an elevation of B_1_ receptor mRNA, without any change of the paw edema caused by the B_2_ receptor agonist tyrosine^8^-BK [33]. Additionally, it was observed that vascular hyporesponsiveness to noradrenaline, in rats that had been pre-treated with LPS, was partially blocked by the selective B_2_ receptor antagonist HOE-140, indicating that part of the hypotensive effects of LPS rely on the activation of B_2_ receptors [34]. The incubation of LPS potentiated BK-B_2_ receptor-mediated signaling pathways, with an increase of intracellular Ca^2+^ and inositolphosphates, in primary cultured tracheal smooth muscle cells. This study points out a relationship between the LPS–TLR4–BK–B_2_ receptor axis in the pathogenesis of bronchial hyperreactivity, by mechanisms involving the activation of Ras-Raf-Mitogen-Activated Protein Kinases (MAPK) [35]. Alternatively, the migration of neutrophils to the lungs of LPS-aerosolized mice was prevented by the ACE inhibitor enalapril, an effect that was reversed by dosing both B_1_ des-Arg^10^-HOE-140 and B_2_ HOE-140 receptor antagonists. In this case, the agonism of either kinin receptor was beneficial to prevent LPS-triggered lung inflammation [36]. More recently, it was demonstrated that des-Arg^9^-BK is a substrate for ACE2 in lungs, and the inhibition of ACE2 might favor the activation of B_1_ receptors by this agonist after LPS inhalation, contributing for neutrophil influx and acute lung inflammation [37]. Moreover, it was demonstrated that BK elicits a biphasic response in LPS-treated rats, with a hypotensive effect followed by a sustained increase in blood pressure, via mechanisms involving a crosstalk between kinin B_2_ and angiotensin-1 receptors [38]. The influence of LPS on kinin-mediated responses point out a relevant role for the kinin system in endotoxemia-related respiratory and cardiovascular complications, besides systemic inflammatory responses, with a special interest concerning septic patients, as discussed in the next section.

### 2.2. The Kinin System in Systemic Inflammatory Response Syndrome (SIRS)

It has been recently reviewed that pathogens have the ability to undermine contact system activation, partly by altering HMWK and BK actions, enabling bacterial dissemination via increased vascular permeability [39]. It was demonstrated that exposure of human lung fibroblasts (IMR-90 cells) to exudates of *Staphylococcus aureus*-treated human peripheral blood mononuclear cells (PBMC) triggered an upregulation of B_1_ and B_2_ receptor mRNA and binding sites. The same protocol of stimulation with *S. aureus*-primed PBMC enhanced the conversion of BK to the metabolite des-Arg^9^-BK. Extending somewhat this experimental evidence, biopsies from patients presenting with staphylococcal soft-tissue infection or erysipelas showed an increased expression of B_1_, but not B_2_ receptors, which had been correlated with enhanced levels of IL-1β at the same sites [40]. Thus, the increased production of des-Arg^9^-BK and the activation of de novo induced B_1_ receptors might be related to *S. aureus* dissemination, facilitating the development of severe conditions. In a mouse model of chronic rhinosinusitis induced by the nasal inoculation of *S. aureus*, there was a marked upregulation of FXII, according to the assessment of the nasal mucosa membrane, suggesting a potential role for BK release in chronification of bacterial sinusitis [41]. FXII has also been described as a plasmatic target for the metalloproteinase CpaA, which is a virulence factor of *Acinetobacter baumannii*, a bacterial species related to antibiotic-resistant ventilator-associated pneumonia and catheter-induced bacteremia. It has been proposed that *A. baumannii* CpaA inhibits the activation of FXII, reducing the host defenses and contributing for the bacterial pathogenicity [42]. The induction of a mouse model of pneumosepsis, by the intranasal administration of *Klebsiella pneumoniae*, triggered a reduction of PPK liver expression. Moreover, the inhibition of PPK by an antisense oligonucleotide reduced both the mortality rates and the infection burden in *K. pneumoniae*-infected mice, likely via activation of innate immune system in lungs [43]. The same research group demonstrated a similar protective effect in the mouse model of *K. pneumoniae* infection, by gene deletion of FXII [44]. However, the modulation of either HMWK or B_1_ and B_2_ kinin receptors failed to provide protection in this experimental paradigm [45,46]. The induction of experimental sepsis by *Streptococcus pyogenes* was associated with a reduction of FXII, kininogen-1 and PPK in mice. Extending preclinical evidence, a decline of FXII and kallikrein levels was observed in plasma of 23 septic patients, irrespective of survival outcomes. Notably, in the mouse model of sepsis elicited by *S. pyogenes*, the inhibition of PPK (but not FXII) markedly prevented the dissemination of *S. pyogenes*, according to assessment in blood and spleen. This strategy also reduced kidney damage and the serum elevation of several cytokines and chemokines, except RANTES (CCL5), which was increased after PPK knockdown. It was concluded that plasma kallikrein blockade might prevent bacterial spread, via inhibition of fibrinolysis [47]. Remarkably, *S. pyogenes* induces BK release from HMWK via streptokinase-activated plasmin, making possible to suggest a relevant role for the kinin system in streptococcal sepsis [48]. Furthermore, gene deletion of HMWK prevented the mortality and reduced endotoxin levels, in a rodent model of sepsis induced by a lethal dose of LPS. Nevertheless, gene ablation of FXII, PPK or B_1_/B_2_ kinin receptors did not provide a protection in this model of SIRS [49].

The participation of the kallikrein–kinin system (KKS) has also been argued in a non-human primate model of sepsis induced by *Escherichia coli* inoculation, indicating that inhibition of FXII, with the consequent reduction of BK release, might prevent septic shock in baboons [50]. The blockade of FXII is able to avoid respiratory complications, intravascular disseminated coagulation, and organ dysfunction, with a decrease of complement cascade activation and cytokine storm. This feasibly contributes to the extended survival rates of septic animals, likely by reducing BK levels and related vasodilation (for review, see [51]). Thus, it is tempting to propose that inhibition of the FXII–kallikrein–kininogen–kinin axis might represent an interesting strategy for preventing septic shock. In this context, a clinical study investigated the inflammatory and coagulation alterations correlating with survival rates in septic children. The authors proposed that prolonged activated partial thromboplastin time is related to poorer survival outcomes, and a reduced activation of FXII might have beneficial effects, probably due to a decline of BK production [52]. Supporting this hypothesis, studies conducted during the 1990s revealed protective effects for selective B_2_ receptor antagonists in preclinical and clinical sepsis. For instance, the administration of NPC17731 prevented the hypotensive responses in a porcine model of sepsis, induced by injection of living *Pseudomonas aeruginosa* [53]. In the clinical setting, the administration of B_2_ receptor antagonist CP-0127 had mild effects on septic patients, with improved survival rates in patients with gram-negative bacterial infections [18]. Compelling pre-clinical evidence also revealed a relevant role for B_1_ receptors in sepsis. Accordingly, the oral treatment with the selective B_1_ receptor antagonist SSR240612 was able to diminish LPS-induced lethality in type 1 diabetic rats by preventing sepsis-related complications such as thrombosis and multiple organ damage [54]. More recently, it was demonstrated that oral treatment with the selective B_1_ receptor antagonist BI113823 was able to reduce systemic inflammation, hemodynamic changes and organ failure, in a rat model of sepsis induced by cecal ligation and puncture [55]. Supporting pharmacological data, transgenic rats overexpressing B_1_ receptors, specifically in endothelium, displayed an increased susceptibility for endotoxic shock [56], similar to what had previously been demonstrated in mice with an overall overexpression of B_1_ receptors [57]. Further studies are necessary to determine how and at what level, the inhibition of the KKS might be useful for reducing mortality in patients undergoing SIRS and septic shock.

### 2.3. Periodontitis and Its Relationship with Kinins

Periodontitis is an inflammatory progressive disease destroying the tooth-supporting tissues, including gingiva, periodontal ligament, and alveolar bone. The interaction between periodontal pathogens and the host leads to a marked inflammatory response, finally resulting in tooth loss. Moreover, periodontal disease has serious impacts on general health, impairing the outcomes of systemic diseases, such as atherosclerosis and diabetes [58]. One of the most important pathogens leading to periodontitis is the anaerobic bacteria Porphyromonas gingivalis. Of note, P. gingivalis LPS mainly activates TLR2 receptors, differently from Escherichia and Salmonella LPS, which typically signal via TLR4 stimulation [59,60]. The high virulence of P. gingivalis is related to the production of cysteine proteinases named gingipains, which cleave LMWK and HMWK, producing Lys-BK and BK in periodontal tissues [61]. Accordingly, it was demonstrated that buccal inoculation of P. gingivalis induced both edema and gingivitis in mice, by the sequential activation of TLR2 and gingipains, leading to the release of kinins and B_2_ receptor stimulation, likely accounting for innate and adaptative immune responses in periodontal disease [62]. Prior evidence demonstrated that gingipain-induced inflammation in the hamster cheek pouch was reduced by the selective kinin B_2_ receptor antagonist NPC 17647 [63]. Furthermore, the increase of vascular permeability induced by P. gingivalis in mice was potentiated by BK or kininase inhibitors, whereas it was hindered by B_2_ receptor antagonism [64]. The intraplantar injection of LPS obtained from P. gingivalis led to an up-regulation of both des-Arg^9^-BK-induced paw edema and B_1_ receptor mRNA in rats, with an involvement of tumor necrosis factor and neutrophil influx [65]. This evidence suggests that kinin receptor modulation might additionally contribute for the pathogenic effects of P. gingivalis in the inflammatory milieu. Reinforcing this notion, a recent publication demonstrated that in vitro activation of TLR2 by P. gingivalis LPS, or the synthetic TLR2 agonist Pam2CSK4, increased the expression of kinin B_1_ and B_2_ receptors in human gingival fibroblasts, at both mRNA and protein levels. Interestingly, the upregulation of either kinin receptor was mimicked in vivo, after the injection of P. gingivalis LPS into the mouse gingiva [66]. It was proposed that gingipains might also play a part for the increased risk of oral cancer in patients with periodontitis—this was associated with the inactivation of SPINK6, an inhibitor of kallikrein activation in the epithelium, probably resulting in augmentation of kinins, which are recognized mitotic mediators [67]. Remarkably, it was found that formation of complex oral biofilms between P. gingivalis and Candida albicans is dependent on the action of gingipains and peptidylarginine deiminase, which in turn leads to kinin citrullination, changing the affinity for kinin receptors, what finally contributes for pathogen escaping mechanisms [68]. Altogether, literature evidence clearly highlights a relevant role for kinins and their receptors in the pathogenesis of periodontitis. Strikingly, periodontitis has negative impacts on metabolic diseases, cancer, and neurodegeneration—all of these conditions being themselves related to a shift of kinin responses. This confirms the kinin system to be a valuable target for treating chronic oral infections and the related complications.

### 2.4. Kinins and Tuberculosis

Tuberculosis is an infectious disease caused by bacillus Mycobacterium tuberculosis, mainly affecting the lungs. According to the Global Tuberculosis Report [69], it is the leading cause of death due to a single infectious agent, affecting 10 million people worldwide. In 2018, approximately 1.2 million TB deaths were estimated among HIV-negative people, with an additional 251,000 deaths among HIV-positive individuals. TB treatment requires the long-term use of a combined scheme of antibiotics, but the numbers of multiple drug-resistant TB cases has been dramatically increased during the last decades. The vaccination protection by M. bovis bacillus Calmette-Guérin (BCG) is partial, primarily preventing the severe forms of TB in children [69]. This indicates the current need to identify new strategies against TB infection. The relationship between Mycobaterium sp. and the kinin system has been sparsely explored. For instance, the systemic administration of BCG, as an inflammatory stimulus, triggered a long-term increase of rat paw edema induced by the selective kinin B_1_ receptor agonists des-Arg^9^-BK and Lys-des-Arg^9^-BK. This protocol of treatment also resulted in a synergistic inflammatory response, by the combination of des-Arg^9^-BK and BK, injected into the rat paw [70]. A similar treatment with BCG also potentiated the nociceptive and edematogenic responses induced by des-Arg^9^-BK, co-injected with formalin into the mouse paw [71]. In both cases, the effects of BCG were impeded by the treatment with the glucocorticoid dexamethasone, indicating an induction of B_1_ receptors. In both rats and mice, the treatment with BCG did not modify the inflammatory or painful responses caused by the selective B_2_ receptor agonist tyrosine^8^-BK, suggesting a selective modulation of B_1_ receptors by M. bovis. In a recent publication [72], it was proposed that kallikrein 12 (KLK12) exerts a protective role against M. bovis infection. This suggestion was based on in vivo and in vitro evidence showing that intranasal inoculation of M. bovis in mice, or its incubation with RAW264.7 macrophages, triggered an upregulation of KLK12. Additionally, KLK12 knockdown reduced the autophagy, apoptosis, and cytokine production in M. bovis-infected macrophages, favoring an increase of bacillus survival. 

As for M. tuberculosis, a clinical study enrolled 13 patients that were submitted to first-line anti-TB therapy, with an induction phase of two months, including isoniazid, rifampin, ethambutol and pyrazinamide, followed by a consolidation phase with isoniazid plus rifampin. Notably, BK and des-Arg^9^-BK serum levels showed variations according to the phase of treatment: whereas BK contents were reduced at the induction phase, des-Arg^9^-BK levels were elevated. Instead, the serum contents of des-Arg^9^-BK were reduced during the consolidation phase of treatment. Thus, kinin levels might represent useful biological markers to predict the outcomes of TB treatment [20]. By using an in vivo model of *M. tuberculosis* infection, it was demonstrated that splenomegaly increased in B_2_ receptor knockout mice, whereas B_1_/B_2_ double knockouts presented a reduction of spleen weight [73]. Gene deletion of either B_1_ or B_2_ receptor did not alter the bacterial load in spleen or lungs. Notably, *M. tuberculosis*-infected wild-type mice displayed an upregulation of B_1_ receptors in spleen and lungs, according to assessment by immunohistochemistry [73]. Finally, the oral treatment with the selective non-peptide B_1_ receptor antagonist SSR240612 reduced the colony-forming units (CFU) in spleen and lungs of *M. tuberculosis*-infected mice. The inhibitory effects of SSR240612 on CFU counts were mirrored in vitro, in *M. tuberculosis*-infected RAW 264.7 macrophages [73]. Further in-depth studies are required to better understand whether the kinin system is involved in *M. tuberculosis*–host interactions, in either latent or active TB forms.

## 3. Viral Infections and Kinins

Viruses are the most common infectious agents worldwide. Although the number of viruses currently known to be pathogenic to humans is relatively small—approximately 220—epidemiological studies strongly suggest that this number is underestimated [74]. In this section, we review some key findings on how viruses modulate host cell biology to facilitate their own replication and how the kinin system impacts viral infections, with a particular focus on Covid-19, hantavirus, and rhinovirus. This emerging field of research provides insights for discovering new anti-viral and anti-inflammatory therapies for managing viral complications.

### 3.1. Crosstalk between SARS-CoV-2 and the Kinin System

Severe acute respiratory syndrome coronavirus 2 (SARS-CoV-2) is a highly contagious virus that was first identified in China (Wuhan city) and rapidly spread worldwide, causing the pandemic coronavirus disease 2019 (COVID-19). SARS-CoV-2 is a member of *Betacoronavirus* family, which includes severe acute respiratory syndrome coronavirus (SARS-CoV) and Middle East respiratory syndrome coronavirus (MERS-CoV), i.e., those responsible for the outbreaks in 2002 and 2012, respectively. SARS-CoV-2 is an RNA virus that is enveloped and anchored by the spike glycoprotein (S) [75,76]. Current evidence shows that the virus invades target cells (e.g., pulmonary endothelial cells, pneumocytes type II, cardiomyocytes, blood cells) through the interaction of spike proteins with ACE2, leading to viral replication [77,78]. The common symptoms of COVID-19 are fever (98%), dry cough (76%), dyspnea (55%), myalgia and/or fatigue (44%), and only a minority of patients show gastrointestinal problems. Approximately 32% of COVID-19 patients are admitted to intensive care unit (ICU) due to respiratory problems—mainly breathing difficulty—with altered coagulation parameters (elevation of D-dimers concentration) [79].

It has been hypothesized that dysregulation of BK-related pathways could be present in Covid-19 patients, being especially related to respiratory complications. Firstly, the SARS-CoV-2 spike antigen binds to ACE2, which in turn internalizes, and thereby downregulates the expression and function of ACE2. Subsequently, there is an increase of blood pressure and pulmonary edema that might evolve to angioedema, likely through generation of BK active metabolites, such as des-Arg^9^-BK. In addition to the modulation of renin-angiotensin system (RAS), des-Arg^9^-BK binds to B_1_ receptors and enhance inflammation and vascular permeability, which is associated with acute lung injury [21]. The inflammatory setting evoked by the des–Arg^9^–BK–B_1_ receptor axis is related to a cytokine storm, with a marked increase in the production of several pro-inflammatory cytokines and chemokines (e.g., TNF, IL-1β, IL-6, CXCL5, CCL2 and CXCL1), finally mediating organ dysfunction [37].

Veerdonk and colleagues (2020) proposed that blocking plasma kallikrein activity (via lanadelumab), together with antiviral treatment, could prevent the acute respiratory distress syndrome in Covid-19, avoiding the ICU admission and mechanical ventilation. In particular, lanadelumab is a monoclonal antibody against the plasma kallikrein, which in turn is responsible for BK production through HMWK precursor, consequently interfering in coagulation cascade and complement system activation [80]. It is important to note that not every patient with Covid-19 will require an inhibition of the kinin signaling, whereas part of the viral load will be resolved in the lungs and thus there will be no second inflammatory wave.

In summary, different approaches could be tested to reduce SARS-CoV-2-associated symptoms and disease exacerbation, including (i) blockade of tissue and/or plasma kallikrein production, (ii) kinin degradation through recombinant ACE2, and (iii) modulation of B_1_ and B_2_ receptors and the downstream pathways.

### 3.2. Hantavirus Infection and BK Signaling

Hantavirus (HV) is a single-stranded, negative-sense viral RNA, belonging to the *Bunyaviridae* family, that causes hemorrhagic fever with renal syndrome (HFRS) and hantavirus pulmonary syndrome (HPS). The HV genus can be divided in two viral lineages: Old World and New World. The first is responsible for HFRS, which is widespread throughout Asia and Europe. The second strain is found in the Americas and is associated with the HPS. Additionally, the main pathophysiological impact in HV infection is systemic vascular leakage, i.e., the patients show impaired vascular tone and increased vascular permeability. All of these features can lead to hypotension, edema, shock, and coagulation abnormalities [81,82]. Preclinical studies have demonstrated that HV firstly invades endothelial cells for replication, and subsequently infect epithelial and vascular smooth muscle cells [83,84]. In line with this, BK promotes important outcomes on the vasculature, such as vasodilation and enhanced vascular permeability. Moreover, the BK–B_2_ receptor axis modulates a variety of second messengers, including intracellular Ca^2+^, endothelium-derived hyperpolarization factor, prostacyclin and nitric oxide, consequently leading to vascular smooth muscle cell relaxation [85]. Indeed, Taylor and coworkers (2013) demonstrated that incubation of FXII, PPK, and HMWK plasma protein, with HV-infected endothelial cells, produces enhanced secretion of BK and reduction of barrier function. In contrast, cells treated with the selective pseudopeptide B_2_ receptor antagonist HOE-140 (icatibant) exhibited a protective activity, with cells maintaining the endothelial integrity. In this study, the real-time cell permeability was assessed using electric cell-substrate impedance sensing [86]. Interestingly, in a case report, a 67-year-old female patient with severe HV infection was treated with HOE-140 (Icatibant, 30 mg, subcutaneously) and after 48 h, she presented a clear improvement of HFRS-related symptoms [87]. Moreover, Antonen and colleagues (2013) showed that HV caused severe capillary leakage syndrome in a 37-year-old Finnish male patient. On the other hand, when the patient received a single subcutaneous dose of the B_2_ receptor antagonist HOE-140 (30 mg), the condition was stabilized followed by gradual improvement. Most importantly, the patient survived. The rationale behind the clinical effects is that the drug can interfere in CD8^+^ T cell-promoted vascular leakage and cytokine secretion, as well as preventing complement activation [88]. It is tempting to suggest that antagonists of B_2_ receptors should be assessed in a broad spectrum of patients with HV infection, mainly in severe cases, i.e., those with life-threatening disease.

### 3.3. Interplay between Rhinovirus and Kinin Pathways

Rhinovirus (RV) is a microorganism popularly known as the “common cold virus”, which is responsible for up to half of acute upper respiratory infections in children and adults. In addition, RV infections also significantly contribute to asthma exacerbations, having a considerable economic burden due to work absenteeism. RV is a member of the family *Picornaviridae* and is a non-enveloped spherical virus, with a diameter of approximately 30 nm [89]. RV has been shown to potentiate the formation of pro-inflammatory mediators, especially of cytokines, via a NF-κB-dependent mechanism [90]. Indeed, mast cells, neutrophils, and airway macrophages secrete histamine, IL-8, and TNF after RV infection, leading to bronchial hyperresponsiveness [91]. Furthermore, Shelfoon and coworkers (2016) demonstrated that RV-16-inoculated human bronchial epithelial cells produced CXCL10 and CXCL8, triggering the migration of fibroblasts, with a subsequent progression of airway remodeling [92].

It was observed that severe infections by RV, mainly those affecting the upper respiratory system, are associated with the enhanced levels of kinins (i.e., BK and Lys-BK) in nasal secretion [93,94]. Alternatively, preclinical studies reported that kallikrein is also detected in lower airways in an allergic-induced asthma experimental model [95]. In 2008, Christiansen and colleagues supported these previous results by showing that tissue kallikrein is markedly activated on days 4 and 18 after RV infection, through evaluation of bronchoalveolar lavage fluid in patients with asthma. The response was accompanied by an increase of IL-8 levels, which was measured by ELISA assay [96].

dsRNA is produced by RV for replication during airway epithelium infection. As mentioned above, several observations suggest that RV infection promotes the local production of kinins. To verify whether dsRNA-derived RV enhances the expression of kinin receptors and exacerbates inflammatory response, Bengtson and coworkers used the human airway epithelial cell line BEAS-2B with dsRNA (Poly I:C) stimulation. Through RT-PCR and radioligand binding, it was possible to verify that B_1_ and B_2_ receptor levels increased in BEAS-2B cells, after Poly I:C administration, which was paralleled with p-ERK activation [97]. The key points of interference within the kinin system that might be useful for management of viral infections are depicted in Figure 2.

### 3.4. Viral Infections and Coagulation–KKS Axis

In the past, researchers were mainly focused on the relationship between bacteremia and the coagulation system. The current focus is on understanding its role in viral infections. Indeed, the coagulation cascade is activated during viral infections, for example by the Ebola virus, HIV, viral pneumonia (H1N1), and emerging pathogens such as Zika and Dengue virus. At first, the response is triggered as a form of protection for the organism, i.e., the immune cells together with platelets are responsible for stopping the spread of the pathogens within the body. Furthermore, fibrin (also called Factor Ia) serves as a scaffold for cell migration and adherence [98]. It is important to state that neutrophils secrete neutrophil extracellular traps (NETs) after tissue factor activation. NETs (e.g., nuclear DNA, histones plus elastase) have been shown to have a potential antiviral effect, mainly due to their negative charge [99]. In contrast, if the coagulation system is dysregulated, the viremia can initiate a disseminated intravascular coagulation and microvascular thrombosis–evoked hypoxia, promoting multiorgan failure and mortality. Hemorrhage can also occur through platelet dysregulation and consumption of coagulation factors, resulting from ongoing intravascular activation of the hemostatic system [100]. FXII can activate the coagulation cascade through FXI, or stimulate the KKS, magnifying the inflammatory responses. Enveloped viruses, such as herpes simplex virus, have been shown to increase the activation of intrinsic coagulation pathway [101]. The relevance of coagulation cascade and KKS has also been explored in COVID-19 [80]. In addition, it has been hypothesized that KKS mediates protection against H1N1 infection. On the other hand, Antoniak and colleagues (2016) demonstrated that there is no difference between HMWK-deficient and wild-type mice in terms of survival parameters, indicating that FXII-dependent KKS activation is not necessary for H1N1 protection. It seems that FXII function is independent on FXI and the KKS stimulation, which could explain the higher mortality in FXII^−/−^ mice in H1N1 infection [102].

## 4. Kinins and Their Implication in Protozoan and Parasite Infections

### 4.1. Kinins in Leishmaniasis Pathophysiology

Leishmaniasis is caused by the protozoans of the genus *Leishmania*, belonging to the *Trypanosomatidae* family [103]. The parasites infect humans through sand-fly female bite. The extracellular forms of the parasite (promastigotes) can infect skin macrophages, transforming into the intracellular amastigote form and invading other macrophages, causing cutaneous leishmaniasis (CL). The parasite also infects mononuclear cells in the circulation and spreads to important organs such as the spleen, liver, bone marrow and lymph nodes in the intestine, leading to the potentially fatal form of visceral leishmaniasis (VL) [104]. Leishmaniasis affects tropical climate territories in Africa, Asia, Americas, and Europe. As of 2018, VL and CL were endemic in 77 and 89 countries, respectively [105]. Parasitological, immunological, and molecular approaches have been used for the diagnosis of leishmaniasis [106]. 

During blood-feeding, the *Leishmania* species vector *Phlebotomy duboscq* deposits saliva in the host dermis, which contains parasites and PdSP15 proteins that inhibit the activation of the contact pathway, preventing BK formation [107]. One clinical study indicated a role for kininogens in the pathophysiology of leishmaniasis, with elevated urinary levels of kininogens in 45 out of 50 kala-azar patients before treatment, suggesting that kinin precursors might be important markers of disease evolution [108]. An in vivo study reported that kinin system is activated by *Leishmania donovani* and *L. chagasi* infection. Hamster cheek pouch was infected topically by promastigotes, inducing macromolecular leakage and inflammatory edema, through the proteolytic release of kinins. The microvascular leakage was ameliorated by pre-treatment with the ACE inhibitor captopril. This response was inhibited by the peptide antagonist HOE-140 or by pre-treatment of promastigotes with K11777, an irreversible cysteine proteinase inhibitor (*N*-methylpiperazine-urea-Phe-homoPhe-vinylsulfone-benzene) [109].

In relation to rodent models, mice with gene deletion of kinin B_2_ receptors displayed a diminished resistance to visceral leishmaniasis [110]. The contribution of B_2_ receptor activation for development of host resistance can be explained by the modulation of the inflammatory response. Svensjo and cols. (2006) showed that captopril pretreatment was able to exacerbate the edematogenic responses in BALB/c or J129 wild-type mice in a paw edema model induced by promastigote injection, whereas HOE-140 blocked this effect. Interestingly, promastigotes were unable to induce edema in captopril-treated B_2_ receptor knockout mice. The in vitro step of the same study reported that B_2_ receptor activation modulated the uptake of promastigotes by macrophages. The treatment with BK at low (5–10 nM) and high concentrations (100–300 nM) augmented and decreased the uptake of promastigotes of *L. chagasi*, respectively, in captopril-treated macrophages. In both cases, this modulation relied on the activation of B_2_ receptors. The exposure of macrophages to K11777- or HOE-treated promastigotes led to a deficient internalization of the parasite through a mechanism dependent on protozoan-produced cysteine proteinases. Even with the reduction of parasite internalization by resident macrophages, the treatment with HOE-140 caused an increase in the rate of growth of intracellular amastigotes, making host cells susceptible to infection, likely by suppressing the protective response of macrophages [109].

It has been suggested that phagocytes are less efficient in internalizing *L. major* expressing inhibitors of serine peptidases (ISP), according to the evaluation of *L. major* WT promastigotes and *L. major* lineages lacking ISP2 and ISP3 (*L. major* Δ*isp*2/3) [111,112]. The investigators concluded that ISP expression impaired the macrophage phagocytosis, by preventing neutrophil elastase-dependent activation of TLR4 [112,113]. More recently, based on the studies cited above, Svensjo and cols. (2014) demonstrated an interplay between the KKS and the inflammatory response evoked by exposure to *L. major* promastigotes in the hamster cheek pouch model. The authors demonstrated that *L. major* WT and *L. major* Δ*isp*2/3 induced a potent microvascular leakage and leucocyte accumulation via the local kinin release. However, *L. major* Δ*isp*2/3 were 20% less effective to induce transendothelial leakage of plasma, when compared with *L. major* WT promastigotes. The topical application of the B_2_ receptor antagonist HOE-140 diminished *L. major* WT-induced plasma leakage. In addition, the in vitro treatment with HOE-140, or with the peptide B_1_ receptor antagonist des-Arg^9^-Leu^8^-BK, inhibited the phagocytic uptake of *L. major* Δ*isp*2/3 by macrophages and had no effects against *L. major* WT promastigotes. The authors also showed that *L. major Δisp*2/3 is capable of sequestering higher concentrations of kininogens than *L. major* WT, concluding that kininogens taken by *L. major Δis*p2/3 are opsonized and are unprotected and susceptible to proteolytic cleavage by pericellular serine proteases of macrophages. Thus, the protective role of ISPs is notable, preventing the action of phagocytes and the kinin release at the site of interaction between the parasite and the phagocytic cells [114].

### 4.2. Kinins Set the Tone in Trypanosoma Infections

There are two types of human trypanosomiasis that are caused by the protozoan of genus *Trypanosoma* [115]. Chagas disease or American trypanosomiasis is caused by *Trypanosoma cruzi*, and is mainly endemic in Latin America, being transmitted by the major triatomine bugs species, such as *Panstrongylus megistus*, *Rhodnius prolixus*, *Triatoma (T.) brasiliensis*, *T. dimidiata* and *T. infestans*. Currently, countries outside Latin America, such as the United States, have experienced a significant increase of imported cases, with an elevation in the numbers of infected individuals. At least eleven species of triatomine bugs have been reported to transmit Chagas in North America (*T. gerstaeckeri*, *T. incrassata*, *T. indictiva*, *T. lecticularia*, *T. neotomae*, *T. protracta*, *T. recurva*, *T. rubida*, *T. rubrofasciata*, *T. sanguisuga*, and *Paratriatoma hirsute*) [116]. Sleeping sickness, or African trypanosomiasis, is caused by *T. brucei (b.) gambiense* or *T. b. rhodesiense*, affecting western sub-Saharan Africa and eastern sub-Saharan Africa, respectively. In this form, the protozoan is transmitted by the tsetse fly bite [117]. The trypomastigote form arrives the bloodstream after contact with a wound or a mucous membrane of the host. The amastigote form is found in muscle and nerve cells [118]. *P. chinai*, *T. dimidiata*, *T. infestans* and *T. rubrofasciata* salivary triafestin inhibits both hemostatic and inflammatory pathways in the host skin, preventing KKS activation [119,120,121,122].

A very few studies describe the association between the KKS and the T. brucei infection. Kinin levels increased in the blood of volunteers infected with T. b. rhodesiense after 2 days of infection, and just after the maximum parasitaemia was reached (12th day). The kininogen levels were inversely proportional to the kinin levels in these individuals [123]. Kinin concentrations were augmented in urine, plasma, ears, skin and feet of mice, rats, rabbits or cattle infected with trypanosomes [123,124,125]. The tropolysin oligopeptidase was described in African trypanosomes, being able to degrade kinins, consequently abolishing the activation of kinin B_2_ receptors [126].

Regarding Chagas disease, a group of Brazilian researchers reported the involvement of the KKS in T. cruzi infection, in a sequence of scientific publications [17,127,128,129,130,131]. Much has been studied about cruzipain, the most abundant cysteinyl proteinase in T. cruzi, as a potential target for development of new therapies for treating T. cruz infection. Firstly, it was demonstrated that cruzipain might be inhibited by rat T-kininogen, depending on the enzyme isoform [132]. In addition, cruzipain is able to release Lys-BK by proteolysis of LMWK, and it also activates PPK [133]. The kinin release, after invasion of non-phagocytic cells (human primary umbilical vein endothelial cells (HUVECs) or Chinese hamster ovary (CHO) cells overexpressing the B_2_ receptor), by T. cruzi trypomastigotes, was potentiated by captopril treatment and inhibited by HOE-140. T. cruzi invasion was also hindered by treatment with the membrane-permeable cysteine proteinase inhibitor Z-(SBz)Cys-Phe-CHN2, demonstrating that cruzipain action is directly associated with kinin release [134]. Moreover, the kinin release by cruzipain activation was modulated through heparan sulfate treatment [135]. Scharfstein and cols. also proposed that B_2_ receptor-mediated intracellular Ca^2+^ elevation takes part of the strategies of protozoan invasion [136]. Alterations in intracellular Ca^2+^ were previously detected in endothelial cells infected by T. cruzi [137]. More recently, Mijares and cols. (2020) described that BK treatment elevates intracellular Ca^2+^ concentrations in cardiomyocytes from Chagasic subjects [138].

An in vitro study identified that treatment with DL-2-mercaptomethyl-3-guanidino-ethylthiopropanoic acid (MGTA), an inhibitor of kininase I (carboxypeptidase M/N), reduced the infection by T. cruzi in B_1_ receptor-expressing cells. In the same study, the mouse paw edema induced by trypomastigotes was mediated by the activation of constitutive B_2_ receptors and induced B_1_ receptors, in the early (3-h) and late-phase (24-h), respectively. It is important to highlight that edema formation was enhanced by captopril treatment [139]. Considering the immune cell infection, Monteiro and cols. demonstrated that dendritic cells (DCs) are sensors to kinins released by T. cruzi, with an involvement of B_2_ receptors [140]. Firstly, the authors described a cooperative activation of TLR2 and B_2_ receptors for induction of type 1 immunity after T. cruzi subcutaneous injection [141]. Additionally, the interplay between TLR2 and B_2_ receptors was linked with CXCR2 activation. Parasitemia and mortality rates were higher in mice lacking B_2_ receptors that had been infected with T. cruzi. Of note, the parasite mRNA expression was augmented in the heart of B_2_ receptor knockout mice. Additionally, effector T cells (CD4^+^ and CD8^+^) isolated from hearts of mice lacking B_2_ receptors produced diminished IFN-γ levels. The type-1 response impairment was also observed in CD11c^+^ DCs isolated from spleen of B_2_ receptor knockout mice that had been infected by trypomastigotes, with reduced IL-12 production. Collectively, these data suggest that B_2_ receptor activation in DCs has a putative role in T. cruzi infection resistance [142]. Corroborating this notion, an in vitro study reported that captopril enhanced the extent of parasite uptake by monocytes, increasing IL-17 expression by CD4^+^ T cells, via activation of kinin receptors [143].

The interaction between TLR2, CXCR2 and kinin B_2_ receptors was assessed in the hamster cheek pouch and mouse paw edema models induced by trypomastigotes. The investigators showed that trypomastigote infection stimulated the production of CXC chemokines via TLR2 activation by macrophages. TLR2 recognition was related to the activation of both B_2_ and CXCR2 receptors, as the inflammatory response induced by trypomastigotes was blocked by HOE-140 and repertaxin (a CXCR2 antagonist) [144]. The endothelial transmigration by T. cruzi was facilitated by BK and increased by the chemokine CCL2, substantiating the relationship between BK and chemokines in T. cruzi pathogenesis [145].

A link between T. cruzi infection and functional changes of endothelin-1 and kinin activity has been described [146]. Initially, Camargos et al. (2002) showed that ET_A_ receptor activation is correlated with the immune response against T. cruzi infection [147]. Afterward, Andrade and cols. (2012) demonstrated that T. cruzi infection is mediated by an interplay between endothelin (ET_A_R and ET_B_R) and kinin B_2_ receptors. In this work, the antagonists of B_2_ receptor (HOE-140), ET_A_R (BQ-123) and ET_B_R (BQ-788) decreased parasite internalization in smooth muscle cells (HSMCs) infected by T. cruzi, and prevented the leucocyte accumulation in the hamster cheek pouch model, while the exogenous endothelin-1 enhanced T. cruzi uptake by HSMCs [148]. Based on this evidence, D’Orléans-Juste and cols. (2012) raised the importance of investigations about the role of de novo induction of B_1_ receptors by cytokines, in Chagas vasculopathies [149].

Schmitz and cols. (2014) suggested a crosstalk between the complement cascade and the kallikrein–kinin system in *T. cruzi* infection. The inhibition of C5a receptor (C5aR) or B_2_ receptor reduced the plasma leakage in the hamster cheek pouch model induced by trypomastigotes, attenuating the trypomastigote-induced IL-12p40/70 responses in CD11c+ DCs [150]. More recently, this research group demonstrated that *T. cruzi*-induced edema is dependent on mast cell degranulation (upstream) and FXII-mediated generation of BK (downstream). Sodium cromoglycate (mast cell stabilizer), infestin-4 (a specific inhibitor of FXIIa), HOE-140 and bosentan (a non-selective antagonist of ET_A_R/ET_B_R) treatments decreased the intracardiac parasite load in mice inoculated with trypomastigotes [151].

The kinin system involvement in chronic Chagas disease (CCHD) associated with systemic arterial hypertension (SAH) was also analyzed. The authors demonstrated an elevation of kinins and nitric oxide in patients with CCHD-SAH, based on increased plasma and tissue kallikrein and plasma kininase II activities, in addition to decreased kininogen concentrations (both HMWK and LMWK) in these subjects [152].

### 4.3. The Kinin System in Malaria

Malaria is caused by *Plasmodium* parasites, encompassing *Plasmodium falciparum*, *P. vivax*, *P. malariae*, *P. ovale* and *P. knowlesi*. The disease has a high prevalence, reaching 228 million cases in 2018, mainly affecting Africa; the *P. falciparum* species is responsible for most cases [153]. The transmission occurs through the bites of infected female *Anopheles* mosquitoes. The symptoms include fever, headache and chills, and if not treated can lead to death [153]. Remarkably, salivary hamadarin and anophensin proteins from *A. stephensi* inhibit BK release and the subsequent host inflammatory reactions [154,155].

In the 1970s and 1980s, researchers held some pieces of evidence about the relevance of kinins in *Plasmodium* infections. HMWK and LMWK levels decreased, while serum kallikrein concentrations were elevated in monkeys infected by *P. knowlesi*. These results were accompanied by increased parasitemia after 3 days of infection and were correlated with inflammatory responses in malaria disease (for review see: [156,157]). In guinea-pig injected intradermally and subdurally with active kallikrein fractions from *P. knowlesi*-infected monkeys, there was an increase of leucocyte infiltrates and endothelial permeability in skin and brain, respectively [158,159]. There was also an elevation of serum kininase activity in infected monkeys, demonstrating the importance of balance among the different components of the kinin system, minimizing the endothelial damage [160]. Ohtomo and Katori (1972) reproduced these data in mice infected by *P. berghei*. The authors observed that severe cases of malaria infection results in parasitemia elevation, diminished plasmatic kininogen and increased kinin formation, with development of hypotension [161]. The increase of kinin levels and kininase activity in the acute stages of malarial infection were correlated with circulatory disturbances [162]. More recently, the release of kinins by *P. chabaudi* and *P. falciparum* parasites was analyzed. In this study, the parasites internalized HMWK and liberated vasoactive kinins (Lys-BK, BK, and des-Arg^9^-BK) through the activation of cysteine proteases falcipain-2 and falcipain-3. The B_1_ and B_2_ receptor activation triggered intracellular Ca^2+^ increase in HUVEC, while the B_1_ and B_2_ receptor antagonists Des-Arg^9^[Leu^8^]-BK and HOE-140 restored this effect [19]. Additionally, Cotrin and cols. (2013) described kininogenase activity for falcipain-2 and falcipain-3 [163].

An in vitro assay reported that BK and its P2 and RI-BbKI analogs presented anti-plasmodial activity against *P. gallinaceum* sporozoites, but not to *P. falciparum* [164]. Another study demonstrated a crosstalk between B_2_ and Mas receptors in invasion of human erythrocytes by *P. falciparum*. BK and Ang-(1–7) impaired the erythrocytic cycle of *P. falciparum*, via PKA activity inhibition, an effect that was inhibited by the selective B_2_ and Mas receptor antagonists, HOE-140 and A779, respectively [165]. Interestingly, kinins have been correlated with placental malaria pathogenesis—*P. berghei*-infected erythrocytes display an upregulation of B_2_ receptors, triggering BK binding and receptor internalization, with erytrocyte engulfment by the trophoblast. In this case, alterations of the BK–B2 receptor axis are likely related to placental dysfunction in malaria [166]. Moreover, the activation of either B_1_ or B_2_ receptor by kinins released from *P. falciparum*-infected erythrocytes modulates the adhesion of infected erythrocytes to endothelial cells, disrupting the blood-brain barrier integrity [167]. In an in vivo study, *P. chabaudi* infection was related to an upregulation of B_1_ receptors in endothelial cells of sinusoids and other blood vessels of mouse liver. Of note, the anti-malarial drug chloroquine modulated the expression of B_1_ receptors in this tissue [168]. A clinical study showed the activation of coagulation cascade in severe malaria caused by *P. falciparum*. The patients presented a reduction of plasma antithrombin III concentrations, an elevation of thrombin-AT 111 complexes and neutrophil elastase, accompanied by a decrease of FXII and PPK activities. The authors suggested that neutrophil elastase and BK influence the pathophysiology of malaria [169]. The role of FXII-PPK was confirmed by Isawa et al. (2007). The investigators showed that anophensin, a KKS inhibitor identified in the salivary glands of vector *Anopheles stephensi*, inhibited kinin production, hindering the activation of both FXII and PPK [155].

### 4.4. Kinins and Snail Fever

Schistosomiasis is a parasitic disease caused by blood flukes (trematode worms) of *Schistosoma* genus; in 2018, it was endemic in 78 countries [170]. Schistosomiasis life cycle occurs by asexual reproduction in snails and sexual reproduction in mammals [171]. Snails release larval forms in freshwater, and these can penetrate skin, infecting individuals. The larvae turn into adult worms that live in the blood vessels inside of body. Eggs released by females, if not expelled, provoke immune reactions and organ damage. Four main species (*S. intercalatum*, *S. japonicum*, *S. mansoni* and *S. mekongi*) and *S. haematobium* are trematodes that infect humans and provoke intestinal and urogenital schistosomiasis forms, respectively [170]. Schistosomiasis is a public health problem that severely affects children and pregnant women, and disease complications can lead to increased morbidity and mortality [172].

Clinical data suggest the participation of kinins in schistosomiasis pathophysiology. PPK and HMWK concentrations were lower in hepatosplenic schistosomiasis patients, when compared with the control group [173,174]. A therapeutic approach employing a short human kininogen insertion into *S. japonicum* glutathione S-transferases, ameliorated their biological activity, and inhibited human umbilical vein endothelial cell proliferation [175].

A study identified a serine protease from *S. mansoni* cercariae genomic DNA: a kallikrein-like protease [176]. The same research group isolated the serine protease SmSP1 from *S. mansoni*, which presents homology with mouse plasma kallikrein and human factor I light chain. This protease likely plays a pivotal role in evasion to the host immune response. Surprisingly, *S. mansoni* infection did not produce specific antibodies to recombinant SmSP1 [177]. In another work, the investigators isolated the sK1 enzyme from *S. mansoni* adult worm homogenate supernatant, which presents kallikrein-like activity. sK1 is present on the surface of adult male worms causing hypotension in a kinin-dependent manner and facilitating the worm movement into the host visceral vasculature. The reduction of arterial blood pressure was demonstrated by intravenous injection of 3 µg of sK1 in rats [178]. As well, BK and its fragment 1–5 were associated with *S. mansoni* schistosomula navigation within the host tissues [179].

Ranasinghe and cols. (2015) identified a gene encoding a single domain Kunitz type protease inhibitor, *SjKI-1*, a coagulation inhibitor from *S. japonicum* that can be used for treatment of hematological disorders, being able to inhibit plasma kallikrein. These authors described *SmKI-1* expression in *S. mansoni,* referring to anti-inflammatory and anti-coagulant properties. Kunitz proteins also protect the schistosomes from the host defense [180]. More recently, the ability of adult schistosomes to cause changes in the mouse plasma proteome was analyzed. In this work, the schistosomes generated carboxyl-truncated forms of HK in comparison with control (murine plasma without parasites), which produces the full-length protein. Schistosomula or adult male worms cleaves HMWK through the action of two cysteine proteases belonging to the calpain family (SmCalp1 and SmCalp2), independent on the activation of kallikreins or BK formation [181]. An anti-hemostatic function was observed to SmSP2 serine protease from *S. mansoni*. The SmSP2 releases BK from kininogen and modulates the host vasodilatation and fibrinolysis, through plasmin activation [182].

In an animal study, an intradermal injection of *S. mansoni* cercariae extracts into guinea pig skin induced edema formation, dependent on local BK production and leukocyte accumulation, while HOE-140 decreased the edema response. The authors also showed that BK did not induce significant leukocyte accumulation [183]. BK administration, in the presence or absence of PGE_2_, before cercarial extracts infection, significantly enhanced edema responses in mouse skin [184]. The therapeutic properties of BK on schistosomal hepatic fibrosis were analyzed in schistosoma cercariae-inoculated mice. The authors demonstrated a reduction of hepatic fibrosis in BK-treated groups, via RAS blockage, with decreased collagen deposition and TGFβ1 protein expression [185]. The role of kinin system in parasite infections is illustrated in Figure 3.

## 5. Kinins and Fungal Infections

*Candida albicans* is the most common opportunistic fungal pathogen in humans and is responsible for up to 50% of invasive candidiasis, albeit it is commensally present in skin and mucosal surfaces (i.e., oral cavity, genitourinary system, and gastrointestinal tract). Additionally, non-*albicans* species have been recognized as serious infection agents in humans, such as *C. glabrata*, *C. parapsilosis*, *C. tropicalis*, *C. krusei*, *C. kefyr*, *C. lusitanie*, and *C. dubliniensis* [186]. In general, immunocompetent individuals do not show *Candida*-related infections. In contrast, subjects with a compromised immunity can display annoying and painful infections, including oral thrush, cutaneous candidiasis, and/or vulvovaginitis. In severe cases, i.e., immunosuppressed patients (HIV, cancer therapy, organ transplantation, or diabetes), the development of systemic candidiasis can occur, being associated with high morbidity and mortality rates [187]. It is important to highlight that the conditions mentioned above are dependent on geographical location, population susceptibility, as well as the use of antifungal drugs.

In this context, kinin production is enhanced in infection as a host defense against invading pathogens, including the recruitment of neutrophils or monocytes to the infection site, or promotion of the pro-inflammatory cytokine secretion through other immune cells. Yet, kinins can exert a dual role, i.e., can be beneficial to pathogens by promoting the increase of vascular permeability, which is necessary for the microorganism nutrition or to colonize the host tissue (Figure 4) [22]. Therefore, tight control over the system is essential to define its activity. Candidadepsin is an extracellular aspartic proteinase that is present in *Candida spp.* and is the major virulence factor of yeasts. Candidadepsin has been proposed to activate kinin production through two routes: (i) the modulation of FXII, and (ii) the enzymatic cleavage of kininogen. Indeed, Kozik and colleagues (2015) showed that LMWK, rather than HMWK, is a substrate for *C. albicans* Sap3 (secreted aspartic protease 3) at sites of infection to produce an amount of kinins [188]. These results were also observed for *C. parapsilosis* Sap1 and Sap2, and all recombinant *C. albicans* Saps, except for Sap7 [189,190]. An approach using proteomics has been used to survey the HMWK-binding proteins, related to *C. albicans* cell wall [191]. Seweryn and coworkers (2015) extended the studies to FXII and PPK, showing that candidal cell wall interacts with these components, mainly five proteins (Als3, Eno1, Eft2, Tpi1 and Gpm1) [192]. The affinity of FXII, PPK and HMWK to fungal protein is evidenced by the works cited above and could be used as potential targets for candidiasis management. More recently, it was demonstrated that BK or kallikrein 1 overexpression protected IL-17 receptor A-knockout mice from candidiasis, by improving kidney function and survival rates. The authors also demonstrated that combination of BK with the antifungal drug fluconazol, resulted in increased survival rates of mice infected with *C. albicans* [193]. Thus, kinin-based strategies might be useful in patients with systemic yeast infections.

## 6. Conclusions

From the pioneer studies showing the relevance of the kinin system in *Plasmodium* infections in the 1970s, passing through the elegant contributions by Dr. Regoli about the effects of bacterial LPS on de novo induction of kinin B_1_ receptors in the 1980s, we arrived in the 1990s with a massive development of kinin antagonists, reaching the 2000s with the approval of icatibant for the treatment of hereditary angioedema, finally meeting the year of 2020 with startling evidence about the relevance of ACE2 and kinins for Covid-19 burden. For every infectious agent—bacteria, virus, fungus, protozoan or helminth—the mechanisms of invasion and escaping from the immune system involve one or more components of the kinin system. A huge number of preclinical studies and some pieces of clinical evidence clearly indicate that pharmacological modulation of kinin precursors, active peptides, kinin receptors and/or related enzymes might provide benefits for managing infectious diseases, including life-threatening situations. Despite this, the number of approved drugs targeting the kinin system is still low. Definitely, there is an urgent need for new options to treat infectious diseases, and the reasons involve the emerging resistance of pathogens and the high toxicity of most antimicrobial drugs, besides unmet clinical needs, such as complex cases of Covid-19. It is about time to reassess the available molecules and to develop novel tools targeting the kinin system as part of the endeavors against infectious diseases.

## Figures and Tables

**Figure 1 pharmaceuticals-13-00215-f001:**
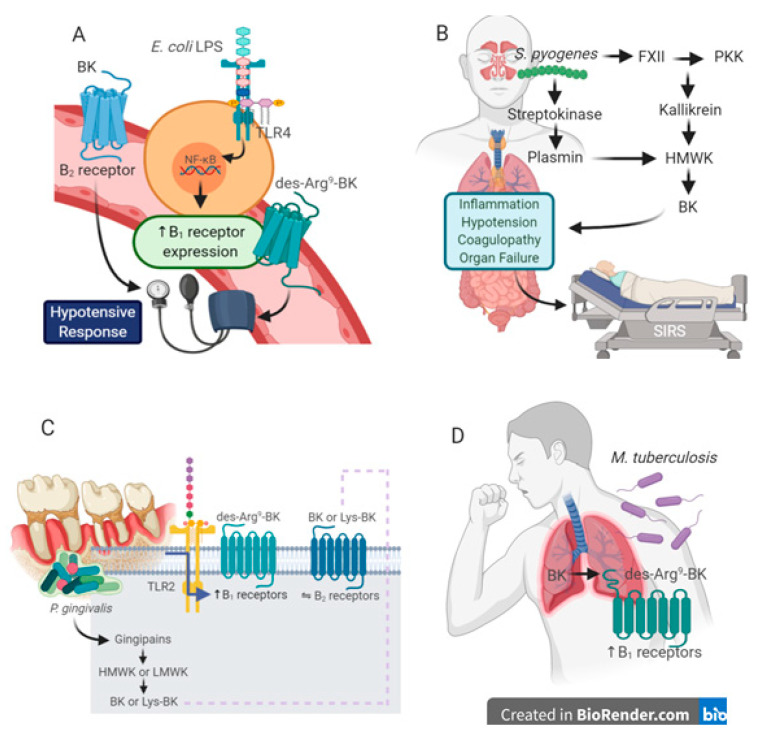
Relevance of components of the kinin system in bacterial infections. (**A**) *E. coli* LPS is a classical stimulus for kinin B_1_ receptor induction in vascular tissues, likely contributing to hypotension in systemic inflammation. *E. coli* LPS-induced B_1_ receptor upregulation rely on TLR4 activation and de novo protein synthesis, mediated by the transcriptional factor NF-κB. Part of the hypotensive effects of LPS are related to an increased production of BK and B_2_ receptor activation. (**B**) Streptococcal respiratory infections caused by *S. pyogenes* trigger the activation of contact system, via modulation of FXII, PPK and kallikrein. Plasmin activation by *S. pyogenes* streptokinase increases BK levels. Upon the bacteria spread throughout the blood stream due to increased vascular permeability, the patient develops systemic inflammatory response syndrome (SIRS) with coagulopathy and hypotension, causing multiple organ failure and shock. The modulation of one or more components of kinin formation pathways might be useful for management of septic patients. (**C**) *P. gingivalis-*produced gingipains cleave kininogen precursors, leading to BK and Lys-BK formation in periodontal tissues. Part of tissue destruction is mediated by the activation of constitutive B_2_ receptors. *P. gingivalis* LPS activates TLR2 and induces an upregulation of B_1_ receptors, which might sustain the chronic inflammation in periodontal disease. (**D**) *M. tubercul*osis infection is able to upregulate kinin B_1_ receptors in lungs, with an increased formation of the selective B_1_ receptor agonist, des-Arg^9^-BK from BK. Pharmacological inhibitors of B_1_ receptors might be therapeutic adjuvants for reducing TB infection burden.

**Figure 2 pharmaceuticals-13-00215-f002:**
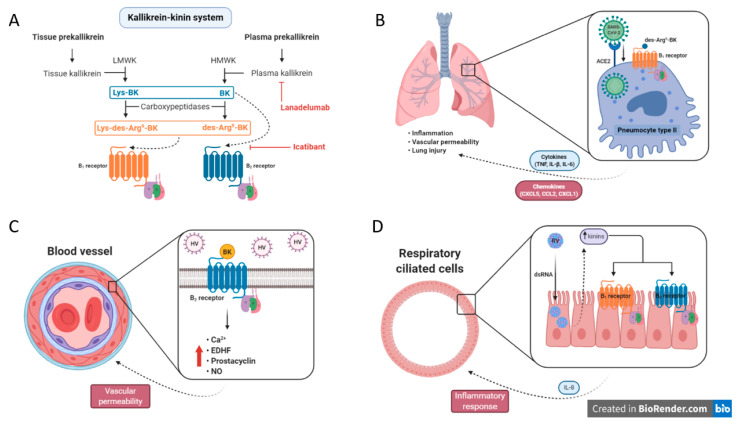
The role of kinin system in viral infections. (**A**) The kallikrein–kinin pathways and their intermediates, including enzymes, active metabolites, and receptors. In addition, the current pharmacological targets to modulate the system (e.g., lanadelumab and icatibant). Dashed lines designate the ligands and their respective receptors. (**B**) Hypothetical mechanism of crosstalk between SARS-CoV-2 and the kinin system, in which it could be one of the responsible routes for the worsening of Covid-19 clinical evolution. (**C**) Hantavirus (HV) modulates the bradykinin (BK)–B_2_ receptor axis promoting the release of Ca^2+^, endothelium-derived hyperpolarization factor (EDHF), prostacyclin and, nitric oxide (NO). Consequently, increasing the vascular permeability. (**D**) The “common cold virus”, called Rhinovirus (RV), has been suggested to enhance the expression of BK receptors (B_1_ and B_2_) by increasing the kinin agonists (i.e., BK, kallidin, and des-Arg^9^-BK) in nasal secretion of patients with respiratory syndromes. LMWK = low molecular weight kininogen; HMWK = high molecular weight kininogen; ACE2 = angiotensin-converting enzyme 2.

**Figure 3 pharmaceuticals-13-00215-f003:**
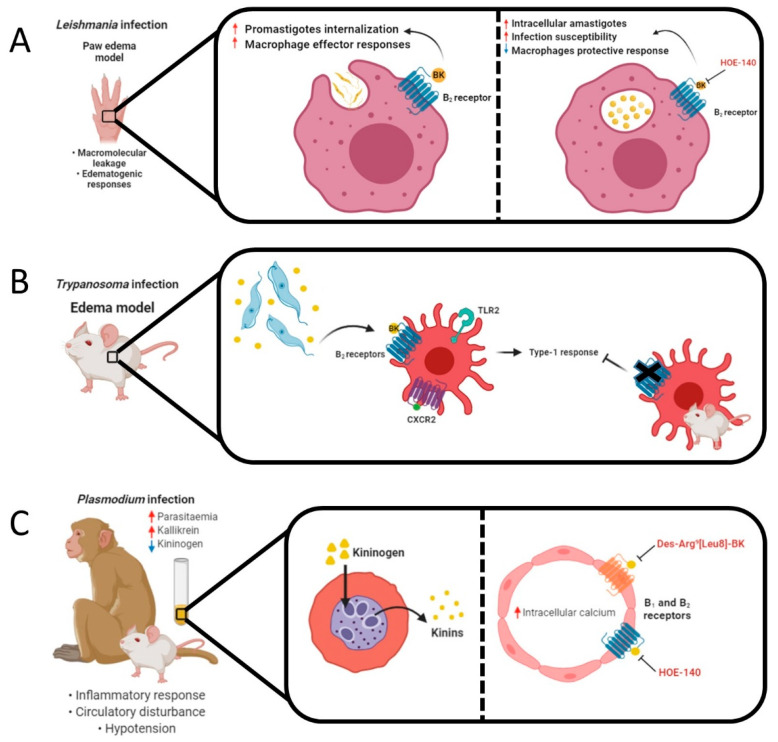
Animal models of parasite infection and the role of kinins. (**A**) *Leishmania* promastigotes induced macrophage effector responses through B_2_ receptor activation by BK in the paw edema model. The treatment with BK increased the uptake of promastigotes by macrophages, whereas HOE-140 blocked this effect. Even with the reduction of parasite internalization, the treatment with HOE-140 caused an increase in the rate of growth of intracellular amastigotes, making host cells susceptible to infection. (**B**) Trypomastigotes release kinins and sensitize dendritic cells (DCs) via B_2_ receptor activation. A cooperative activation of TLR2, CXCR2 and B2 receptors induces type 1 immunity. An impairment of type-1 responses was observed in CD11c+ DCs isolated from spleen of B_2_ receptor knockout mice, that had been infected by trypomastigotes, with reduced IL-12 production. (**C**) Monkeys or mice infected by *Plasmodium* presented decreased HMWK and LMWK levels, while serum kallikrein concentrations and kinin formation were elevated. These results were accompanied by increased parasitemia and kininase activity. Parasites internalized plasmatic kininogen and liberated vasoactive kinins (Lys-BK, BK, and des-Arg^9^-BK) through the activation of cysteine proteases falcipain-2 and falcipain-3. B_1_ and B_2_ receptor activation triggered intracellular Ca^2+^ increase in endothelial cells causing circulatory disturbances, while the selective kinin antagonists des-Arg^9^[Leu^8^]-BK and HOE-140 restored this effect. (**D**) Intradermal injection of *Schistosoma* cercariae into the guinea pig skin induced edema formation, BK release and leukocyte accumulation, while HOE-140 decreased the edema response. Adult male worms cleave HMWK through protease activation. Proteases trigger BK production from kininogen, stimulating the release of tissue plasminogen activator (tPA) from vascular endothelial cells, which would promote fibrinolysis and anticoagulant effects.

**Figure 4 pharmaceuticals-13-00215-f004:**
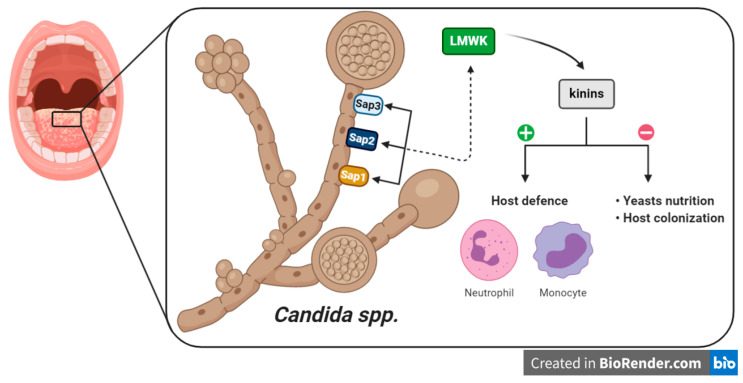
Involvement of kinin system in candidiasis. The pathogenicity of *Candida* spp. in sequential stages of interaction (Sap -> LMWK -> kinins -> effects) with kinin signaling in human host infection. Sap = secreted aspartic protease; LMWK = low molecular weight kininogen.

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
