# Peer review of "Kinins and Their Receptors in Infectious Diseases"

_pharmaceuticals, 2020, doi:10.3390/ph13090215_

Round 1

Reviewer 1 Report

In this manuscript, the authors discuss the role of the kinin system and kinins’ receptors in some infectious diseases.

This is an interesting paper, however, some points need to be reinforced:

The authors need to explain the relationship between kinins and the regulation of hemostasis.

Also, authors need to discuss the dependence of the outcome of infectious diseases on the disbalance of the hemostatic system. Infectious disease including those caused by viruses and bacteria are known to be associated with hemostatic disorders including disseminated intravascular coagulation.

Since the main focus of the article is on receptors, it is necessary to specify the title of the review.

Author Response

Ref.: Manuscript ID: pharmaceuticals-901461

REPLY TO THE COMMENTS TO THE REVIEWER # 1’S REPORT

In this manuscript, the authors discuss the role of the kinin system and kinins’ receptors in some infectious diseases. This is an interesting paper, however, some points need to be reinforced.

We would like to thank this Reviewer for the positive comments and the relevant criticisms about our review article. We have made an effort to suitably address your suggestions in the revised version of the manuscript.

The authors need to explain the relationship between kinins and the regulation of hemostasis.

As recommended, we have inserted a new statement in the Introduction of the manuscript (section 1) for explaining the relationship between kinins and hemostasis regulation (lines 24-26).

Also, authors need to discuss the dependence of the outcome of infectious diseases on the disbalance of the hemostatic system. Infectious disease including those caused by viruses and bacteria are known to be associated with hemostatic disorders including disseminated intravascular coagulation.

Thank you very much for this valuable criticism. Please, note that we have further discussed the relationship between kinins and the disbalance of hemostatic system in virus infections, within the item 3 of the manuscript (new item 3.4; lines 414-436). For bacterial infections, this issue was clarified throughout the item 2.2. of the manuscript (lines 171-182).

Since the main focus of the article is on receptors, it is necessary to specify the title of the review.

According to your recommendation, the paper is now entitled: “Kinins and their receptors in infectious diseases”. Thank you for this suggestion. 

Reviewer 2 Report

The manuscript presented by Dr. Dagnino and colleagues is a comprehensive review with exciting and relevant data, highlighting the importance of kinins in inflammation and their potential use as targets to treat infectious diseases. I highly recommend the acceptance of this manuscript after some minors edits:

  1. Some statements are too vague in phrases where a simple word or two may make the complete full and more understandable. For instance, the sentence
  2. Line 43. The line states that " The pathophysiological roles of kinins and their receptors have been greatly favored by development of peptide and non-peptide selective receptor ligands”. How were these developed? Are these recombinant versions of the ligands? It may seem simple, but fixing these small issues may increase the manuscript's impact reaching a more ample audience.
  3. Also, Kinins are a large family of proteins. I suggest starting the review (first paragraph), naming some of the most prominent members of these family (Bradikinin, for example), where they are synthesized and their mechanism of action as a whole before entering the specifics. This can be done in one or two short paragraphs.
  4. Line 35: "by most cells at 35 peripheral and central levels” Please, give an example of such cells.
  5. Lines 49 – 51: In the paragraph stating that “ Along the last four decades, the role of either kinin receptor and other components of the kinin system has been described in a series of infectious conditions, such as sepsis, schistosomiasis, leishmaniosis, Chagas Disease, candidiasis, tuberculosis, malaria, and it has been recently correlated with Covid-19 infection”, all the citations for these studies are missing.
  6. Line 114. ACE is mentioned at least 11 times in the manuscript. However, there is not a single paragraph detailing the nature of the molecule. Since it is an important molecule to study due to its involvement in SARS-COv-2 pathogenesis, this information is relevant.
  7. Line 234: The authors state that “the systemic treatment with BCG triggered a long-term increase of rat paw edema induced by the selective kinin B1 receptor agonists des-Arg9-BK and Lys-des-Arg9-BK.” Who receives this type of treatment? When is it administrated, and why? This is needed because the previous paragraph only talks about BCG as a vaccine, not as a treatment option. This needs to be clarified for people outside of the field.
  8. Line 408: “can infect skin macrophages, transforming into the intracellular amastigote form and invading other macrophages” Please, complement this paragraph by saying that amastigotes replicate inside the cell, bursting it then infecting new cells causing the lesions or something similar. 
  9. Line 464: Chagas is also endemic in vectors from the United States, and the number of vector species is significantly higher than just one species. This needs to be corrected in the manuscript. Also, please see:

SOSA-ESTANI, Sergio and SEGURA, Elsa Leonor. Integrated control of Chagas disease for its elimination as public health problem - A Review. Mem. Inst. Oswaldo Cruz [online]. 2015, vol.110, n.3 [cited 2020-08-07], pp.289-298. Available from: <http://www.scielo.br/scielo.php?script=sci_arttext&pid=S0074-02762015000300289&lng=en&nrm=iso>. ISSN 0074-0276. http://dx.doi.org/10.1590/0074-02760140408.

Trypanosoma cruzi and Chagas' Disease in the United States

Caryn Bern, Sonia Kjos, Michael J. Yabsley, Susan P. Montgomery

Clinical Microbiology Reviews Oct 2011, 24 (4) 655-681; DOI: 10.1128/CMR.00005-11 

10. Line 591. It would be useful to talk a little bit about the sialokinin or other kinin potentially contained in saliva of arthropod vectors of human and animal disease. can they also have an impact in the vertebrate skin immunity?

11. Line 668. I suggest making fungal infections as a separate section.

Author Response

Ref.: Manuscript ID: pharmaceuticals-901461

REPLY TO THE COMMENTS TO THE REVIEWER # 2’S REPORT

The manuscript presented by Dr. Dagnino and colleagues is a comprehensive review with exciting and relevant data, highlighting the importance of kinins in inflammation and their potential use as targets to treat infectious diseases. I highly recommend the acceptance of this manuscript after some minors edits.

We would like to thank this Reviewer for the encouraging comments about our review article and also by the valuable comments and suggestions that will certainly improve the quality of the paper.

  1. Some statements are too vague in phrases where a simple word or two may make the complete full and more understandable. For instance, the sentence

Please, observe that we have made an effort to improve the sentences as indicated by this Referee. Details of changes are provided below, for each point raised by the Reviewer.

  1. Line 43. The line states that " The pathophysiological roles of kinins and their receptors have been greatly favored by development of peptide and non-peptide selective receptor ligands”. How were these developed? Are these recombinant versions of the ligands? It may seem simple, but fixing these small issues may increase the manuscript's impact reaching a more ample audience.

Thank you very much for this criticism. We have described that development of kinin receptor ligands mainly involves chemical synthesis. We have also included new references in this part of the text . (lines 52-53).

  1. Also, Kinins are a large family of proteins. I suggest starting the review (first paragraph), naming some of the most prominent members of these family (Bradikinin, for example), where they are synthesized and their mechanism of action as a whole before entering the specifics. This can be done in one or two short paragraphs.

Please, note that main kinins, and the mechanisms of kinin formation and degradation are described in the first paragraph of the Introduction (Section 1; lines 22-34)). Additionally, the mechanisms of action are described within the second paragraph of the same section (lines 38-50). In the revised version of the paper, we have made an effort to improve both paragraphs, as recommended by this Reviewer.

  1. Line 35: "by most cells at peripheral and central levels” Please, give an example of such cells.

Please, note that we have included some examples of cells expressing kinin receptors in peripheral and central levels (lines 43-44).

  1. Lines 49 – 51: In the paragraph stating that “ Along the last four decades, the role of either kinin receptor and other components of the kinin system has been described in a series of infectious conditions, such as sepsis, schistosomiasis, leishmaniosis, Chagas Disease, candidiasis, tuberculosis, malaria, and it has been recently correlated with Covid-19 infection”, all the citations for these studies are missing.

We apologize for this. Please, observe that we have cited references regarding the relevance of the kinin system in the different infectious diseases at this part of the text (line 61; references 17-22).

  1. Line 114. ACE is mentioned at least 11 times in the manuscript. However, there is not a single paragraph detailing the nature of the molecule. Since it is an important molecule to study due to its involvement in SARS-COv-2 pathogenesis, this information is relevant.

Your comment is totally relevant. Please, note that we have included a new statement in the Introduction (section 1; first paragraph; lines 34-37) about the main features of ACE and ACE2.

  1. Line 234: The authors state that “the systemic treatment with BCG triggered a long-term increase of rat paw edema induced by the selective kinin B1 receptor agonists des-Arg9-BK and Lys-des-Arg9-BK.” Who receives this type of treatment? When is it administrated, and why? This is needed because the previous paragraph only talks about BCG as a vaccine, not as a treatment option. This needs to be clarified for people outside of the field.

In this case, we have used BCG as an inflammatory stimulus to further characterize the upregulation of kinin B1 receptors by infectious agents. Please, note that we have made an effort to clarify this point in the revised version of the paper (lines 242-243).

  1. Line 408: “can infect skin macrophages, transforming into the intracellular amastigote form and invading other macrophages” Please, complement this paragraph by saying that amastigotes replicate inside the cell, bursting it then infecting new cells causing the lesions or something similar.

Your comment is totally relevant. Please, note that we have complemented this statement accordingly (lines 442-443).

  1. Line 464: Chagas is also endemic in vectors from the United States, and the number of vector species is significantly higher than just one species. This needs to be corrected in the manuscript. Also, please see:

SOSA-ESTANI, Sergio and SEGURA, Elsa Leonor. Integrated control of Chagas disease for its elimination as public health problem - A Review. Mem. Inst. Oswaldo Cruz [online]. 2015, vol.110, n.3 [cited 2020-08-07], pp.289-298. Available from: <http://www.scielo.br/scielo.php?script=sci_arttext&pid=S0074-02762015000300289&lng=en&nrm=iso>. ISSN 0074-0276. http://dx.doi.org/10.1590/0074-02760140408.

Trypanosoma cruzi and Chagas' Disease in the United States  Caryn Bern, Sonia Kjos, Michael J. Yabsley, Susan P. Montgomery Clinical Microbiology Reviews Oct 2011, 24 (4) 655-681; DOI: 10.1128/CMR.00005-11

We apologize for that missing information. As recommended, we have improved this part of the review, by naming the other vector species implicated in Chagas Disease, besides describing that Chagas Disease is also endemic in the USA, besides Latin America countries. The suggested references have also been included in the revised paper (lines 500-506).

  1. Line 591. It would be useful to talk a little bit about the sialokinin or other kinin potentially contained in saliva of arthropod vectors of human and animal disease. can they also have an impact in the vertebrate skin immunity?

Thank you very much for this consideration. Sialokinins are tachykinin analogues (such as substance P) detected in the saliva of Aedes aegypti mosquitoes, although they are not related to the BK-family of peptides. However, some vectors implicated in the diseases described in our manuscript are able to release kinin inhibitors from the saliva, what is likely associated with the mechanisms of successful infection. Therefore, we decided to include this information set in the revised version of the manuscript (lines 449-451; 510-512; 595-597).

  1. Line 668. I suggest making fungal infections as a separate section.

Your comment is totally pertinent. As required, we presented fungal infections as the new section 5, of the revised version of the paper.
